# Causal reasoning over knowledge graphs leveraging drug-perturbed and disease-specific transcriptomic signatures for drug discovery

**Daniel Domingo-Fernández**[1]*, **Yojana Gadiya**[1], **Abhishek Patel**[1], **Sarah Mubeen**[2], **Daniel Rivas-Barragan**[3], **Chris W. Diana**[1], **Biswapriya B. Misra**[1], **David Healey**[1], **Joe Rokicki**[1], **Viswa Colluru**[1]*

**1** Enveda Biosciences, Boulder, Colorado, United States of America, **2** Bonn-Aachen International Center for IT, Rheinische Friedrich-Wilhelms-Universität Bonn, Bonn, Germany, **3** Barcelona Supercomputing Center, Barcelona, Spain

* daniel.domingo-fernandez@envedabio.com (DDF); viswa.colluru@envedabio.com (VC)

**Data Availability Statement:** All links to relevant data are within the manuscript under the

## Abstract

Network-based approaches are becoming increasingly popular for drug discovery as they provide a systems-level overview of the mechanisms underlying disease pathophysiology. They have demonstrated significant early promise over other methods of biological data representation, such as in target discovery, side effect prediction and drug repurposing. In parallel, an explosion of -omics data for the deep characterization of biological systems routinely uncovers molecular signatures of disease for similar applications. Here, we present RPath, a novel algorithm that prioritizes drugs for a given disease by reasoning over causal paths in a knowledge graph (KG), guided by both drug-perturbed as well as disease-specific transcriptomic signatures. First, our approach identifies the causal paths that connect a drug to a particular disease. Next, it reasons over these paths to identify those that correlate with the transcriptional signatures observed in a drug-perturbation experiment, and anti-correlate to signatures observed in the disease of interest. The paths which match this signature profile are then proposed to represent the mechanism of action of the drug. We demonstrate how RPath consistently prioritizes clinically investigated drug-disease pairs on multiple datasets and KGs, achieving better performance over other similar methodologies. Furthermore, we present two case studies showing how one can deconvolute the predictions made by RPath as well as predict novel targets.

## Author summary

Different types of interactions between various biological elements (e.g., proteins, drugs and diseases) can be modeled using networks for various applications, including drug discovery and finding novel use cases of known drugs. Nevertheless, we are far from having a complete picture of all possible biological interactions that can occur in humans, and so,

Implementation Details section and can be found at
https://github.com/enveda/RPath.

**Funding:** DDF, YG, AP, CWD, BBM, DH, JR, and
VC have been funded by Enveda Biosciences. This
work has been funded by Enveda Biosciences
(https://www.envedabio.com/). The funders had no
role in study design, data collection and analysis,
decision to publish, or preparation of the
manuscript. SM and DRB received no specific
funding for this work.

**Competing interests:** I have read the journal's
policy and the authors of this manuscript have the
following competing interests: DDF, YG, AP, CWD,
BBM, DH, JR, and VC are employees of Enveda
Biosciences Inc. during the course of this work and
have real or potential ownership interest in Enveda
Biosciences Inc.

current networks modeling human biology remain incomplete. To try and compensate for this shortcoming, researchers are beginning to use both knowledge of biological interactions, alongside experimental data. In this work, we show how we can deduce which drugs may be good candidates for treatments by using networks to estimate how a drug can affect a disease, and overlaying elements in our network with those in experimental datasets. These experimental datasets can help guide us through the network, showing us which interactions are likely occurring and which are not. Finally, we show that the approach we take can also help us to come up with new research questions and determine which proteins a drug must actually target to produce a therapeutic effect in a patient.

This is a *PLOS Computational Biology* Methods paper.

## Introduction

The representation of biomolecular interactions occurring within cells is often intuitively organized in the form of biological networks. These networks can be used to inherently model biological processes through the use of nodes denoting biological entities and edges representing their relationships. While homogeneous networks, such as protein-protein interaction networks, can represent relationships between a single entity type, knowledge graphs (KGs) can incorporate a broad range of biological scales, from the genetic and molecular level (e.g., proteins, drugs, and biochemicals), to biological concepts (e.g., phenotypes and diseases). These KGs can then be utilized for several applications in drug discovery, such as providing insights into molecular mechanisms and therapeutic targets [1–2], side effect prediction in the early stages of drug development [3], target prioritization [4], and drug repositioning [5].

Given the flexibility of KGs, multiple heterogeneous relation types can be modeled to represent biological processes that are governed by interactions occurring between component entities [6]. Even though a variety of relation types (e.g., literature co-occurrence, associations, etc.) can be leveraged by network-topology algorithms for various applications, causal relations are particularly useful as they can be used to infer the effect of any given node on another by reasoning over the KG [7]. Nonetheless, not all interactions included in a given KG are necessarily biologically relevant as they may be context-specific, such as to a particular cell type, tissue or disease. Furthermore, as the complete human interactome remains unknown, KGs modeling PPIs are also incomplete and the interactions which are modeled tend to be biased towards well-studied proteins and their relationships [8]. One approach to address these challenges is to jointly leverage prior knowledge in KGs with data-driven *-omics* experiments [9–13].

Experimental datasets have been widely employed by recent drug repurposing approaches to identify drug candidates for a given disease using the anti-correlation in biological processes or pathways at the transcriptomic- or proteomic- level between drugs and diseases as a proxy [13–16]) (see [17] for a recent review and **S8 Table** for a survey of such methods). While these approaches use prior knowledge in the form of pathways (gene sets), this concept has yet to be applied on KGs for drug discovery. However, by mapping the signatures of an *-omics* experiment to a KG, we can not only verify which causal interactions are observed within a specific context, but also prioritize and identify the mechanism of action of a drug for a given disease with high precision.

Currently, there exist numerous algorithms that leverage causal relations for the interpretation of -*omics* data. In general, these algorithms operate by assessing the concordance between transcriptomic or proteomic signatures and the predicted causal effects encoded in these relations [18–19]. For instance, the Reverse Causal Reasoning (RCR) [20] and Network Perturbation Amplitude (NPA) algorithms [21–22] assess and score this concordance employing causal graphs consisting of up-stream and down-stream proteins (nodes) representing regulations occurring in biological pathways. Subsequently, the scores obtained from these algorithms can be used for the interpretation of -*omics* data commonly derived from contrast experiments. Although the interpretations obtained from these algorithms may be relevant for several downstream applications, such as drug target prediction, disease characterization, and side effect prediction, the algorithms themselves cannot be directly used for these applications. Additionally, these algorithms have been specifically designed for bipartite graphs, thus, simplifying biological pathways to a single relation between two proteins.

While traditionally, these algorithms were applied on small causal networks, they have recently begun to be applied on large-scale KGs, given the increasing availability of causal information, including proteins, drugs and phenotypes. For instance, a recent algorithm we published, drug2ways, reasons over all paths between a drug and a disease in a KG to predict the effect of the drug as the cumulative effect of all directed interactions between these two nodes [23]. Reasoning over all paths overcomes the limitation of earlier algorithms that exclusively account for shortest paths on protein-protein interaction networks, oversimplifying the effect exerted by one node on another, as all other paths between the two nodes are ignored [24–25]. Nonetheless, paths in large-scale KGs can grow exponentially, many of which may not be relevant in a true biological context. Thus, incorporating signatures from context-specific experimental datasets along with prior knowledge in a KG can enable us to reason over the entire network-structure and ensure only paths which can be observed in a biologically meaningful context are retained. In doing so, we can address several of the limitations of the above-mentioned methods for drug discovery.

Here, we present RPath, a novel algorithm that prioritizes drugs for a particular disease by reasoning over causal paths in a KG, guided by both drug-perturbed and disease-specific transcriptomic signatures (**Fig 1**). We demonstrate how RPath is able to recover a large proportion of clinically investigated drug-disease pairs on multiple transcriptomic datasets and KGs, performing better than other network-based methods. Furthermore, we show two additional applications where we illustrate how our approach can also assist in hypothesis generation and target prioritization.

## Results

This section is divided into three subsections that outline the different applications of RPath presented in this manuscript. First, we demonstrate how RPath can be used to identify potential drug candidates for various diseases using a variety of KGs and datasets, outperforming numerous link prediction methods. Next, we leverage the inherent interpretability of KGs to generate hypotheses for the predictions made by RPath. Finally, we outline how RPath can be reversed-engineered and alternatively used to predict targets for a given disease.

### Identification of drug candidates

To demonstrate the ability of our algorithm to accurately identify drug candidate for a given disease, we evaluated its performance to recover clinically investigated drug-disease pairs using two distinct KGs and four transcriptomic datasets (i.e., two each containing numerous

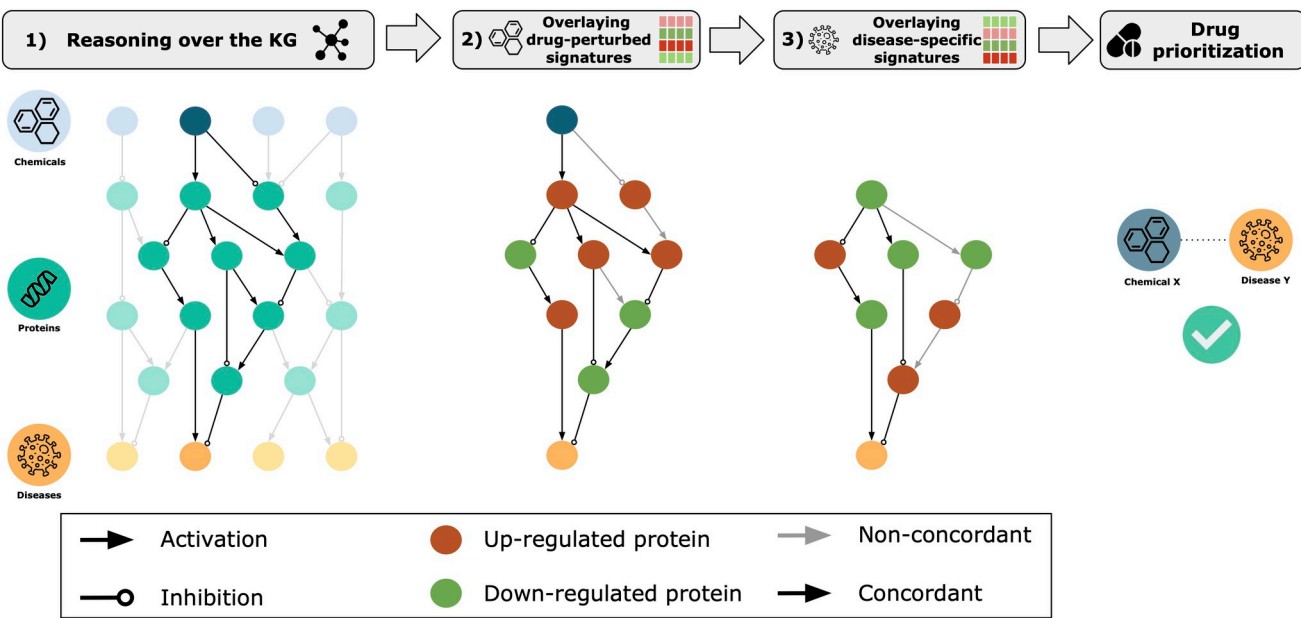

**Fig 1. Schematic representation of the RPath algorithm. Step 1)** All acyclic paths of a given length between a drug and a disease in the KG are calculated. If there exist causal acyclic paths connecting the drug and the disease, a subgraph involving all these paths is inferred. This subgraph represents the proposed mechanism of action by which the drug may be a therapeutic target of the given disease. **Step 2)** Transcriptomic signatures observed from a drug-perturbed experiment are overlaid onto each corresponding node present in these paths. Then, RPath traverses through each path and evaluates whether the inferred direction of regulation (i.e., activation or inhibition) at every step is concordant with the up- and down- regulations (i.e., red and green nodes, respectively) observed in the transcriptomic signatures. **Step 3)** In a similar manner, transcriptomic signatures observed within a specific disease context are overlaid onto each corresponding node in the concordant paths from the previous step (if any). Next, RPath evaluates whether the disease transcriptomic signatures contradict the paths that were concordant with the drug signatures. If this is the case, the specific drug-disease pair is prioritized.

drug-perturbed and disease transcriptomic experiments). In this task, RPath consistently prioritized a significantly larger number of clinically investigated drug-disease pairs across all datasets and in both KGs compared with the precision expected by chance (i.e., probability of randomly picking a positive label among drug-disease combinations that are connected through a path) (**Table 1**).

The highest precision values were found for the L1000-GEO datasets with 80% and 66.67% for the OpenBioLink and custom KGs, respectively. In the remaining datasets, the precision was approximately 50%, except for the CREEDS-Open Targets datasets in the custom KG that exclusively yielded a single drug-disease pair which was not in clinical trials. While the precision expected by chance approximately varied between 10% and 42%, RPath consistently achieved higher precision values across nearly all datasets and KGs, ranging between 50% and 80% (e.g., more than five times higher for the L1000-GEO dataset in OpenBioLink running

**Table 1. Evaluation of RPath in multiple datasets across the two KGs using precision.** Each row corresponds to the results of running RPath on a specific drug-disease dataset combination. The second and fourth columns show the performance that is expected to be achieved by chance.

| - | OpenBioLink KG | | Custom KG | |
|---|---|---|---|---|
| Dataset combination | Precision (TP/TP+FP) | Expected precision by chance | Precision (TP/TP+FP) | Expected precision by chance |
| **L1000 [26]–GEO [27]** | 80% (4/5) | 17.42%% | 66.67% (2/3) | 13.74% |
| **L1000 –Open Targets [28]** | 54.55% (6/11) | 15.01% | 50% (2/4) | 9.62% |
| **CREEDS [29]–Open Targets** | 50% (1/2) | 32.66% | 0% (0/1) | 24.40% |
| **CREEDS–GEO** | 50% (1/2) | 41.15% | 50% (1/2) | 34.08% |

RPath (80%) vs. chance (17.42%)). Notably, the number of prioritized drug-disease pairs were constrained for two reasons: i) RPath requires transcriptomic information for a given drug and disease and, ii) the pair must also be present in the KG (**see S1 Table for details**). Furthermore, apart from the low number of drug-disease pairs that fulfilled these criteria, RPath filters the majority of pairs with paths between them after overlaying the transcriptomic signatures in Steps 1 and 2 (**see Fig 1**) of the algorithm (**S2 Table**). For example, in the case of the CREEDS-GEO datasets and the OpenBioLink KG, the total number of diseases was 10, resulting in only a couple of drug-disease pairs being prioritized. Nonetheless, we were still able to validate our methodology across multiple datasets and KGs, observing that RPath performed significantly better than chance at identifying clinically investigated drug-disease pairs.

Finally, we benchmarked RPath against 11 alternative methods [30–31] that have been used to predict drug-disease links in a KG with the same characteristics as the ones used in this work. The precision of these methods varied between 5% and 43% (**S3 Table**). Furthermore, since the majority of these methods prioritize a drug and a disease based on their network proximity (e.g., shortest paths and number of shared nodes), these methods recurrently prioritized the same set of drug-disease pairs. Thus, these methods could not be used to prioritize drugs outside the vicinity of disease-associated proteins since only a minority of drug-disease pairs are connected by a single protein, but the majority of them contain longer paths that are not considered by these methods (**S2 Table**). Lastly, we also conducted permutation experiments where we permuted both the binarized gene expression values (i.e., +1, -1, 0) observed in the transcriptomic datasets and the edges of the KGs, while maintaining their underlying structure. The results of our experiments showed how the number of prioritized drug-disease pairs significantly decreases when permuted datasets and KGs are employed and that none of these few prioritized pairs were clinically investigated (**S9 Table**).

## Interpretation of the mechanisms of action of the proposed drug candidates

In a case study, we sought to explore the results obtained by running RPath on the custom KG. Of the prioritized drug-disease pairs (see **S3 Text**), we studied the paths between two of the pairs to demonstrate how our approach can potentially be used to deconvolute the mechanism of action of some drugs (**Fig 2**). We selected bicalutamide and ponatinib as these two anti-cancer drugs were the top-ranked prioritized drugs and already approved for prostate cancer and acute myeloid leukemia, respectively. Furthermore, since the mechanisms of action of these drugs have been widely studied, we can compare the mechanistic paths identified by RPath against known interactions and pathways reported in scientific literature.

First, we investigated ponatinib, a multi-targeted tyrosine-kinase inhibitor, which is used to treat acute myeloid leukaemia (AML) (**Fig 2B**). Among the targets of this drug present in the concordant paths for this pair, we were able to identify fms-like tyrosine kinase 3 (FLT3), which is mutated in approximately 20% of AML patients [32] and several members of the FGFR family proteins. Furthermore, we observed other proteins including KDR, LYN, and SRC, all of which are kinase-associated targets in AML. As a downstream target of these proteins, we found JAK2, a well-studied player in myeloproliferative diseases, with known mutations and hypermethylation events. We further identified the transcription factor, CEBPA, that is critical for normal development of granulocytes and is also implicated in AML [33] and the SPI1 gene, from which circSPI1, a circular RNA derived from the gene, has recently been shown to be highly expressed in AML patients [34]. Other proteins that are inhibited as a result of the signaling cascade triggered by ponatinib include KIT, which is implicated in cell death in AML [35]. RAS family members NRAS and KRAS, both of which are associated with the

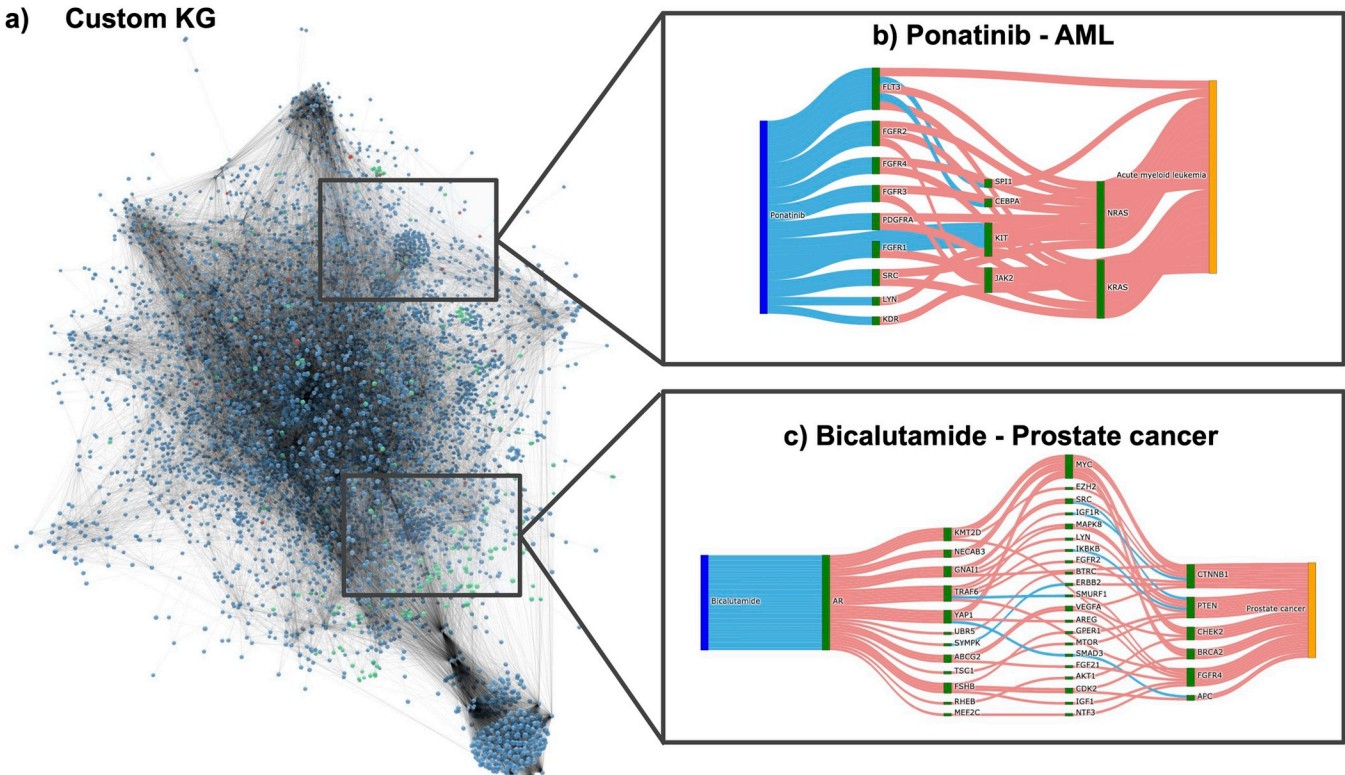

**Fig 2. Devoncoluting the mechanism of action of a drug through RPath.** By investigating all the paths of a given length between a drug and a disease in a KG, we can analyze the different mechanisms that are proposed by RPath. **a)** Visualization of the custom KG. Proteins are colored in blue, diseases in red and drugs in green. Sankey diagram illustrating a sample of the paths between ponatinib and AML (**b**) and bicalutamide and prostate cancer (**c**) for the custom KG. Activatory relations in the Sankey diagrams are colored in red and inhibitory relations in blue.

prognosis of solid tumors and hematological malignancies, including AML [36] were also implicated.

The second studied drug-disease pair is bicalutamide, used for the treatment of prostate cancer. Bicalutamide is an anti-androgen medication that binds to the androgen receptor (AR), as illustrated in **Fig 2C**. The paths between bicalutamide and prostate cancer point to several downstream targets of this drug, including the epigenetic regulator KMT2D, which is known to sustain prostate carcinogenesis by epigenetic mechanisms [37], and NECAB3, known to enhance the activity of HIF1A, thus promoting glycolysis under normoxic conditions and enhancing tumorigenicity in cancer cells [38]. Furthermore, we were able to identify CTNNB1, which plays a role in the development of numerous prostate cancers [39]. Interestingly, we also observed novel players that have not yet been reported in the literature, such as GNAI1, SYMPK, UBR5, and MEF2C that may provide new insights on the mechanism of action of this drug.

## Target prioritization

Prior to the identification of a therapeutic drug candidate for any given disease, a crucial first step is often to identify biologically relevant protein targets. Ideally, the perturbation of a particular protein target in a disease state should result in the reversal of the observed phenotype. In a similar manner to the above-mentioned applications, by reasoning over the KG guided by disease signatures, RPath can be used for target prioritization. Since, as per our knowledge,

**Table 2. Top 5 prioritized protein target-disease pairs.** These results were obtained by running RPath over both KGs with the GEO and Open Targets datasets using the same path length as the drug discovery task (see **Methods**). Pairs were prioritized based on the number of concordant paths. The vast majority of pairs were prioritized using the disease transcriptomic signatures from the GEO dataset given its larger coverage of measured genes compared to Open Targets (**S4 Table**).

| Protein target | Disease | Concordant paths | Nodes in the concordant paths | KG | Transcriptomic dataset |
|---|---|---|---|---|---|
| NOG | AML | 18,456 | 1,008 | Custom KG | GEO |
| PRKCA | AML | 12,861 | 669 | Custom KG | GEO |
| CXCL8 / IL-8 | AML | 7,234 | 465 | Custom KG | GEO |
| NOG | Plasma cell myeloma | 5743 | 616 | Custom KG | GEO |
| CDC42 | Medulloblastoma | 5,651 | 91 | OpenBioLink | GEO |

there are no large datasets that contain information about known targets for a wide variety of indications, we could not conduct a validation strategy similar to the analyses presented in the subsection *Identification of drug candidates*. Instead, we focused on evaluating the top prioritized protein targets across all diseases using literature evidence (**Table 2**).

Among the top protein target-disease pairs proposed by RPath, two have already been associated with AML, including PRKCA, for which several drugs already exist [40–41] and CXCL8/IL-8 [42–44]. Furthermore, CDC42, which has been proposed as a candidate target for medulloblastoma, plays a role in several cancers. Specifically, CDC42 has been shown to act as a regulator of medulloblastoma-associated genes [45] and compounds for its inhibition have also been proposed [46].

## Discussion

In this work, we present a novel methodology that leverages prior knowledge from causal relations across multiple biological modalities in KGs and assesses their concordance with transcriptomic signatures for drug discovery. In the past, several algorithms have been primarily introduced for the interpretation of transcriptomic signatures by reasoning over shortest paths [24–25] or bipartite graphs [20–22]. Though these algorithms could also be indirectly applied for drug discovery, they present some shortcomings: i) they operate on homogeneous causal graphs with a single entity type (e.g., protein nodes), ii) they are solely conducted on single contrast experiments (e.g., drug-treated vs. control), and iii) they do not fully exploit all possible paths in these causal graphs. RPath addresses these shortcomings by reasoning over all possible causal paths in a multimodal KG and leveraging both drug and disease transcriptomic signatures. First, our algorithm reasons over the ensemble of paths between a given drug and a disease in a KG. Second, it evaluates the concordance of these paths against the transcriptomic changes experimentally observed for that drug. Third, it assesses whether the effect of these paths is opposite to the transcriptomic signatures observed within the disease context. In a final step, the algorithm identifies potential drug candidates as those whose paths correlate with drug-perturbed transcriptomic signatures and are anti-correlated to the disease transcriptomic signatures. We have validated our methodology in eight independent analyses, finding that RPath consistently identifies a large proportion of clinically investigated drug-disease pairs over multiple datasets and KGs. Additionally, we conducted several robustness experiments and benchmarked the algorithm against 11 network-based methodologies. Finally, we also showed how our approach can be used to deconvolute the mechanism of action of a drug as well as to prioritize protein targets for a given disease.

We acknowledge a few shortcomings in our work that are worth discussion. Firstly, we were limited by the availability of high-quality annotated transcriptomic datasets for drugs and diseases, as only four of the approximately 30 datasets that we identified met our requirements. Furthermore, the coverage of measured genes varied largely across experiments. For instance,

while the average number of genes measured in the Open Targets dataset was approximately 900, that number dropped to 500 in the CREEDS dataset (**S4 Table**). In contrast, the total number of proteins in the KGs were in the range of several thousands. As RPath requires that signatures from both these drug and disease datasets be mapped to the KG, most of the proteins in the KGs could not be quantified. Thus, we allowed for up to one error when calculating the concordance in the path between a drug and a disease. Furthermore, two other reasons justified an error within the path. Firstly, introducing an error limits the impact of an arbitrary fold change cut-off, which ultimately determines the up-/down-regulation of each protein. Secondly, some paths might contain causal relations that do not reflect a change at the transcription level of the affected protein (e.g., phosphorylation of a protein kinase) [18–19]. We expect that this challenge we faced of quantifying proteins in our KGs will be overcome by high-quality, consistent datasets such as those generated in large pharmaceutical enterprises and emerging data-driven biotech companies looking to leverage large-scale computational technologies. Another characteristic of our approach is that the identification of a potential drug for a given disease requires knowledge of the protein target and the effect of the drug on it. However, this information is not always available or must be inferred using computational approaches. Finally, the interpretation of the mechanism of action of a proposed drug with the help of scientific literature comes with the caveat that the individual interactions were themselves derived from the literature. Nonetheless, it is still possible to interpret the mechanism of action of a drug irrespective of the aforementioned limitation as the paths of the proposed drug-disease pairs include only those which are concordant with observed data-driven transcriptomic signatures.

While we have demonstrated our novel algorithm across multiple datasets and KGs, we envision multiple other applications. Firstly, by incorporating time series data into the analysis, we can determine how the paths between the drug and the disease are altered over time following the concept outlined by [47]. Secondly, although we have demonstrated our methodology using transcriptomic data, other modalities can be used if the KG contains causal relations for these entities (e.g., metabolomics). Additionally, although we have employed transcriptomic signatures in this work, we acknowledge that RNA levels may not directly reflect the functional activity of proteins. However, given the growth in the availability of proteomic data, we envisage the application of our approach on proteomic experiments from databases such as PRIDE [48], ProteomicsDB [49], and L1000 [26] in the future. Furthermore, although a multimodal KG may lack the context within which each relation occurs, RPath inherently takes this into account by removing the paths which do not match the observed transcriptomic signatures. However, the algorithm could also be applied on a disease-specific KG in order to model the pathophysiological mechanisms characteristic of a given phenotype [50–51].

## Methods

### Theoretical background

We denote a KG as a set of nodes and edges, where nodes correspond to three distinct biological entities (i.e., chemicals, proteins, and diseases) connected through causal relations, representing activatory or inhibitory effects. Causal relations within the KG connect drug-protein, protein–protein, and protein–disease nodes. A (directed) path in a KG is defined as a sequence of two or more biological entities connected through causal relations. Paths in the KG can be either cyclic or simple. A cyclic path refers to paths in which one or more nodes repeat, whereas a simple path corresponds to a path in which no nodes appear more than once. The length of a path is defined by the number of edges that connect the nodes within the path.

## RPath algorithm

The algorithm used in our framework, RPath, reasons over the paths in a KG to identify all possible effects a given drug can have on a disease (**Fig 1**). Each of these paths can be divided into three main sequential parts that attempt to represent the mechanism of action of a drug: i) the drug activates/inhibits a protein target (drug-protein edge), ii) the protein target triggers a signaling cascade (a set of protein-protein edges), and iii) the signaling cascade reverts the disease condition (protein-disease edge). Furthermore, since every causal edge contains information on the effect each node exerts on another (i.e., activation or inhibition), we can infer the direction of regulation (i.e., up-/down-regulated) for each node at each step of a path [24–25].

Once the causal acyclic paths between a particular drug and disease in the KG have been calculated (**Fig 1; step 1**), the next step of RPath is to overlay transcriptomic signatures from a drug-perturbed experiment (**Fig 1; step 2**). We hypothesize that because a number of paths might represent the biologically relevant mechanism of action of this drug, the observed transcriptomic signatures for proteins in the KG should be concordant with the inferred up- or down-regulations at every step of the path. For example, if in a given path, a drug inhibits a protein target and that target activates a signaling cascade, we expect the inhibition of the protein target as well as the inhibition of the proteins downstream of the target. We would like to note that a gene is considered to be differentially expressed if its expression is significantly altered with respect to a reference sample (i.e., control). Keeping this in mind, a cut-off is applied to each measured gene in the experimental dataset based on the fold change; this measurement is used to define whether the gene is up-/down-regulated or unchanged.

Similarly, the final step of RPath involves overlaying disease-specific transcriptomic signatures to the nodes in the paths of the KG (**Fig 1; step 3**). We hypothesize that, in contrast to the overlaying of drug-perturbed signatures, transcriptomic signatures in a disease context should be anti-correlated to both the drug-perturbed signatures as well as the inferred up- or down-regulations for every node in the path. This final step is inspired by previous work that exploited the anti-correlation between drug and disease signatures at the pathway level for drug repurposing [15–16]. In summary, RPath aims at prioritizing a specific drug for a given disease if i) there exist causal paths between the drug and disease in the KG, ii) the causal effects on these paths are aligned with the transcriptomic changes observed in the drug-perturbed experiment, and iii) both the drug signatures and the paths are anti-correlated with the transcriptomic dysregulations observed in the disease. **Fig 3** outlines the pseudocode of the described logic of the algorithm.

As an additional application, the algorithm can be modified following the same logic for target prioritization (**see S1 Fig for the pseudocode).** This variant of the algorithm begins from a disease of interest and calculates all paths from the disease to all proteins for a given path length (e.g., a path length of 6). Next, it calculates the concordance between the paths for each potential protein target and the transcriptomic signatures of the given disease to assess whether there are proteins that could be key up-stream regulators of the observed phenotype. We would like to note that this application exponentially increases the running time of the algorithm as it requires querying paths from a disease to several thousands of proteins in the KG, as opposed to only a handful of chemicals.

## Datasets and validation

In this subsection, we present drug-perturbed and disease-specific transcriptomic datasets as well as the KGs used to demonstrate our methodology. We then introduce the strategy we follow to validate our methodology.

---

**Algorithm** Algorithm to prioritize a drug candidate through its correlation to drug transcriptomic signatures and anti-correlation to disease transcriptomic signatures.

```
 1: function IS_DRUG_PRIORITIZED(KG, drug, disease, lmax, errors_allowed)
 2:     paths ← GET_ACYCLIC_PATHS(KG, drug, disease, lmax)
 3:     if |paths| == 0 then
 4:         return false
 5:     end if

 6:     drug_tr ← GET_TRANSCRIPTOMICS(drug)
 7:     disease_tr ← GET_TRANSCRIPTOMICS(disease)

 8:     filtered_paths ← ∅
 9:     for all path ∈ paths do
10:         if IS_CONCORDANT(KG, drug_tr, disease_tr, errors_allowed) then
11:             filtered_paths.insert(path)
12:         end if
13:     end for
14:     if |filtered_paths| == 0 then
15:         return false
16:     end if

17:     anti_correlated_paths ← ∅
18:     for all path ∈ filtered_path do
19:         if IS_ANTI_CORRELATED(path, drug_tr, disease_tr, errors_allowed) then
20:             anti_correlated_paths.insert(path)
21:         end if
22:     end for
23:     if |anti_correlated_paths| == 0 then
24:         return false
25:     end if
26:     return true
```

---

**Function 1** Assess whether the path between a drug and a disease is concordant with the observed drug transcriptomic signatures

```
 1: function IS_CONCORDANT(KG, path, drug_tr, errors_allowed)
 2:     errors ← 0
 3:     change ← 1
 4:     source ← path[0]
 5:     for i ← 1; i < |path|; i++ do
 6:         target ← path[i]
 7:         change = KG(source, target) * change
 8:         if change ≠ drug_tr[target] then
 9:             errors ← errors + 1
10:             if errors > errors_allowed then
11:                 return false
12:             end if
13:         end if
14:     end for
15:     return true
```

---

**Function 2** Assess whether the path between a drug and a disease anti-correlates with the observed disease transcriptomic signatures.

```
 1: function IS_ANTI_CORRELATED(path, drug_tr, disease_tr, errors_allowed)
 2:     errors ← 0
 3:     for all protein ∈ path do
 4:         if drug_tr[protein] == disease_tr[protein] then        ▷ do not anti-correlate
 5:             errors ← errors + 1
 6:             if errors > errors_allowed then
 7:                 return false
 8:             end if
 9:         end if
10:     end for
11:     return true
```

---

**Fig 3. Pseudocode of the RPath algorithm.** Given a KG, drug, disease and a defined path length (i.e., *lmax)*, the core function of the algorithm, *is_drug_prioritized*, returns whether a drug should be prioritized or not. For this, the function calculates all acyclic paths between a drug-disease pair in the KG. For each path found, drug-perturbed (i.e., drug_tr) and disease-specific (disease_tr) transcriptomic signatures are overlaid onto their corresponding protein nodes. The function then prioritizes the drug if at least one path is concordant with the observed drug-perturbed transcriptomic signatures (evaluated via **Function 1**, *is_concordant*) and the same path is anti-correlated with the observed disease-specific transcriptomic signatures (evaluated via **Function 2**, *is_anti_correlated*). Paths which match both the drug-perturbed signatures and contradict disease-specific signatures are then returned by RPath as promising drug candidates.

---

**Drug-perturbed and disease transcriptomic datasets.** We identified four databases that were suitable for our approach (**S5 Table**); drug-perturbed transcriptomic data were obtained from CREEDS [29] and L1000 [26] while disease transcriptomic data were collected from Open Targets [28] and GEO [27]. All experimental datasets from these resources (downloaded on 15.02.2021) contained gene expression changes measured in humans. Drugs and diseases from datasets obtained from these databases were then mapped to PubChem compound identifiers and the Mondo Disease Ontology (MONDO), respectively, for consistency with the entities of the knowledge graphs presented in the next subsection. Similarly, gene identifiers in all datasets were harmonized to ENTREZ. Of the four databases, datasets from L1000 contained a binarized value for the direction of dysregulation for every gene (i.e., up-regulation and down-regulation), while for the remaining databases, fold changes were binarized for significantly dysregulated genes using $|\log_2 \text{fold change}| = 1$ as a cutoff (**S1 Text**). As fold change thresholds tend to be arbitrary selected [52], we opted to select a threshold of 1 as opposed to a more stringent one (e.g., $|\log_2 \text{fold change}| > 2$) to ensure a larger number of dysregulated genes would be retained. Finally, we conducted a systematic search for databases that contained either a large number of drug-perturbed or disease-specific transcriptomic datasets. While this search initially resulted in 27 candidate databases (**see S5 Table for details about each dataset**), the majority of them were not suitable for our study as they either contained too few transcriptomic datasets or the drugs/diseases in these datasets were not in the KGs used to demonstrate our methodology.

**Knowledge graphs.** We demonstrate our methodology using two established publicly available KGs that contain causal relations across drugs, proteins, and diseases: OpenBioLink KG [53] and a custom KG [23]. Both KGs are originally generated from a compedia of independent databases; thus, containing unique causal interactions depending on the source databases they include. As outlined in the algorithm, the KGs are required to encompass three types of causal edges: drug-protein (i.e., drug activates/inhibits protein), protein–protein (i.e., protein activates/inhibits protein), and protein–disease (i.e., protein activates/inhibits disease). Furthermore, the original node identifiers for drugs and diseases in both KGs were respectively mapped to PubChem compound identifiers and MONDO to be consistent with the transcriptomic datasets. Next, we removed drugs and diseases that were not present in any of the four transcriptomic datasets presented in the previous subsection as the paths between these drug-disease pairs cannot be validated. **Fig 4** shows the final statistics of both KGs after the previously outlined filtering steps. **S4 and S6 Tables** summarize the overlap between the genes

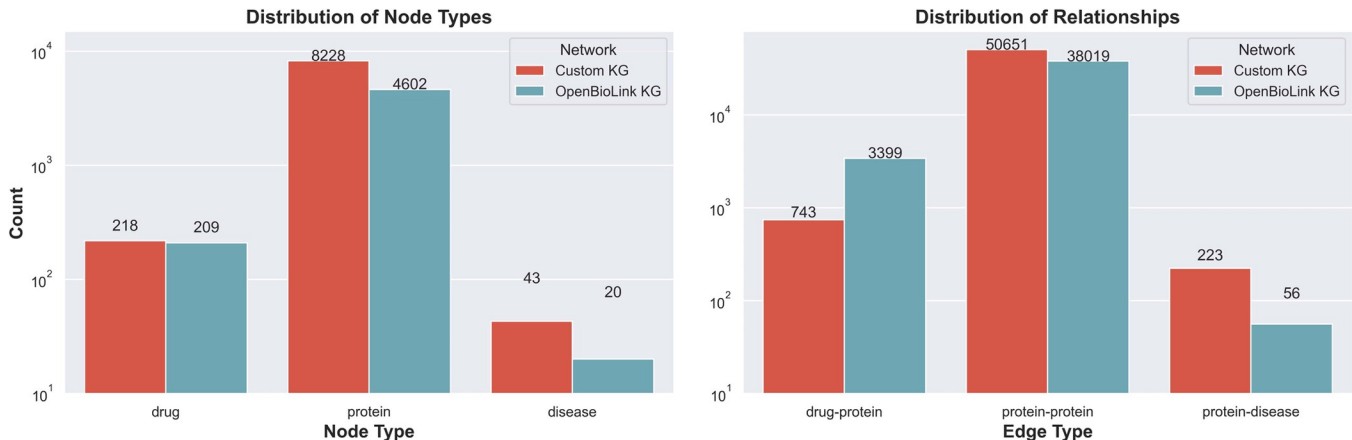

**Fig 4. Distribution of node and edge types in the custom and OpenBioLink KGs.** The properties of each of the two networks are detailed in **S7 Table**.

measured in each of the four transcriptomic datasets and their corresponding protein nodes in the KGs.

**Validation.**   In line with other network-based approaches designed for drug discovery [54–55], we used drug-disease pairs that have been clinically investigated as positive labels, extracting this information from ClinicalTrials.gov (accessed on 28.09.2020). Clinical trials are commonly used as a proxy for highly validated, medically relevant biological interactions, independently of whether the clinical trial was successful or not, as multiple *in vitro* and *in vivo* studies must first validate the interaction in order for the drug to proceed to a clinical trial. Notably, this assumption may result in a worse performance being reported than the actual performance of the algorithm as some of the drug-disease pairs that are considered as negative labels may in fact be positive ones.

Since drugs and diseases in ClinicalTrials.gov are formalized using MeSH identifiers, we harmonized these identifiers to the ontologies used in the KG (i.e., PubChem compounds for drugs and MONDO for diseases) (mappings are available at the GitHub repository). After the harmonization, any drug or disease in the KGs that was not present in any clinical trial or did not have any path to any disease in the KG was subsequently removed, as its corresponding node could not be used in the presented validation. Details on the harmonization procedure are provided in **S1 Table**.

To test the robustness of RPath in identifying these clinically investigated drug-disease pairs, we conducted eight independent analyses, one for each of the combinations of the two drug datasets, the two disease datasets, and the two KGs (e.g., CREEDS-GEO-OpenBioLink, L1000-GEO-OpenBioLink, etc.). For each of these eight analyses, we ran RPath over a given KG to prioritize drug-disease pairs among all possible drug-disease combinations. We would like to note that these pairs prioritized by the algorithm are those whose paths are both correlated with drug-perturbed transcriptomic signatures and anti-correlated with disease transcriptomic signatures. Furthermore, we make two assumptions in the design of the algorithm. Firstly, paths with cycles or a length greater than 7 edges between a given drug and disease are not considered, assuming that the effects exerted by paths beyond this length are less biologically relevant [23]. Secondly, we allow for at most one error between the transcriptomic data and a given path (see pseudocode of the algorithm in **Fig 3**). We refer to an error in the path as the disagreement between the type of causal interaction (i.e., activation or inhibition) and the direction of dysregulation of genes in the transcriptomic datasets (i.e., up or down -regulation), or the absence of the drug-perturbed and/or disease-specific transcriptomic signature. We restricted the number of allowed errors to at most one as, without any errors, running the algorithm over the two KGs with most dataset combinations will not yield any prioritized pairs, and permitting more than one error will result in an exponential increase in the number of prioritized pairs. For example, in the latter case, for a path of length 5 (i.e., a sequence of 3 proteins), an allowance of two errors (e.g., missing expression values for 2 of the 3 proteins) would still result in the prioritization of the drug-disease pair if the remaining protein both correlated with the drug and anti-correlated with the disease, obfuscating results.

From this set of prioritized drug-disease pairs, we expect to retrieve a larger proportion of clinically investigated drug-disease pairs (i.e., positive labels) than expected by chance (i.e., proportion of positive labels in the dataset that also have a path between the drug and disease). Here, it is important to note that, as in any drug discovery task, there is a class label imbalance where the vast majority of the drug-disease pairs are negative labels while the proportion of positive labels ranges from anywhere between 9% and 41% for each of the eight analyses (**S4 Table**). Furthermore, this type of validation falls into the so-called early retrieval problem. In other words, from the thousands of drug-disease pairs that are tested, we are exclusively prioritizing the top-ranked pairs that have been equally prioritized by the algorithm. This small

subset represents the interesting drug-disease pairs that would be further investigated in the drug discovery process. In such cases, it is inadequate to apply metrics such as receiver operating characteristic (ROC) curves as they operate on a full ranked list. Therefore, it does not necessarily evaluate the ability of a model to prioritize the most promising drug-disease pairs candidates [56]. Additionally, considering that not all drug-disease pairs have been clinically studied, a number of the negative labels might be falsely classified as positive. To address these issues, we evaluated the performance of RPath based on the ratio of true positives that appear in the prioritized drug-disease pairs (i.e., precision was used as the performance metric). As a baseline, we assessed whether the prioritized drug-disease pairs found through the algorithm contain a larger proportion of positive labels (i.e., drug-disease pairs investigated in clinical trials) than expected on average by chance.

As a benchmark, we compared RPath against 11 equivalent approaches that can be used to prioritize drug-disease pairs based solely on network structure, as outlined by [26] and [27] (S2 Text). The choice of these approaches is motivated by the fact that, as per our knowledge, there are no network-based methods that operate on multimodal KGs using transcriptomic signatures for the prioritization of drug-disease pairs. Additionally, we conducted a validation experiment where we simultaneously randomly permuted the directionality of the genes measured in the transcriptomic datasets and the KGs using the XSwap algorithm [57] while both preserving network structure and the original gene expression distributions. Using these, we then rerun the eight analyses to compare the significance of our results [57].

## Implementation details

The RPath algorithm and the benchmarked methods are implemented in Python leveraging NetworkX (v2.5) (https://networkx.github.io). Network visualizations were done using WebGL, D3.js, Three.js, Matplotlib and igraph. Source code, documentation, and data are available at https://github.com/enveda/RPath. The validation presented in the paper can be reproduced by running the Jupyter notebooks available at https://github.com/enveda/RPath/tree/master/notebooks.

## Supporting information

**S1 Text. Processing of transcriptomic datasets.**
(DOCX)

**S2 Text. Benchmarked methods.**
(DOCX)

**S3 Text. Prioritized pairs.**
(DOCX)

**S1 Table. Clinical trial information mapped to the OpenBioLink and custom KGs.** For each possible pairing of a drug-disease database (i.e., column 2) with entities that could be mapped to either the OpenBioLink or custom KG, we report the proportion of drug-disease pairs contained in ClinicalTrials.gov (i.e., column 3). We consider these clinically investigated drug-disease pairs as positive labels for the validation of our approach. The number of unique drugs (PubChem compound identifiers) and diseases (MONDO identifiers) are reported in columns 4 and 5, respectively, while the total number of possible combinations of these unique drugs and diseases are presented in column 6.
(XLSX)

**S2 Table. Percentage of drug-disease pairs at different steps for every dataset combination and KG.**
(XLSX)

**S3 Table. Evaluation of the 11 benchmark methods using precision as a metric.** None of the benchmarked methods achieve a precision greater than 50%. Furthermore, we would like to note that most of the drug-disease pairs prioritized by each of these methods are the same since they are based on network proximity. Thus, if a drug and a disease share a large number of nodes, they will consistently be prioritized by most of these methods.
(XLSX)

**S4 Table. Statistics on the genes measured in the four transcriptomic datasets used.**
(XLSX)

**S5 Table. Investigated datasets.**
(XLSX)

**S6 Table. Overlap of the transcriptomic dataset and the KGs.** The total number of drugs and diseases in the datasets which can be mapped to the KG as well as the proportions of them that are present in the KG are given in columns 3 and 4, respectively. Similarly, column 5 displays the total number of mapped proteins as well as the proportions of proteins that are present in the KG. Details about each individual drug/disease are available at https://github.com/enveda/RPath/blob/master/data/drug_disease_overview.tsv.
(XLSX)

**S7 Table. Properties of the OpenBioLink and custom KGs.**
(XLSX)

**S8 Table. Drug repurposing approaches exploiting anticorrelation of transcriptomic signatures.**
(XLSX)

**S9 Table. Permutation experiments across the four dataset combinations using permuted KGs and gene expression datasets.**
(XLSX)

**S1 Fig. Pseudocode of the RPath algorithm designed for target prioritization.**
(TIF)

## Acknowledgments

We are very grateful for the publicly available content provided by several databases which we have used in this work (i.e., the KGs and transcriptomic datasets).

## Author Contributions

**Conceptualization:** Daniel Domingo-Fernández.

**Data curation:** Daniel Domingo-Fernández, Sarah Mubeen, Daniel Rivas-Barragan.

**Formal analysis:** Daniel Domingo-Fernández, Yojana Gadiya, Abhishek Patel, Sarah Mubeen.

**Funding acquisition:** David Healey, Joe Rokicki, Viswa Colluru.

**Investigation:** Daniel Domingo-Fernández, Yojana Gadiya, Abhishek Patel, Biswapriya B. Misra.

**Methodology:** Daniel Domingo-Fernández, Yojana Gadiya, Joe Rokicki.

**Project administration:** Daniel Domingo-Fernández, Chris W. Diana, Biswapriya B. Misra, David Healey, Joe Rokicki, Viswa Colluru.

**Resources:** Daniel Domingo-Fernández, Chris W. Diana, Viswa Colluru.

**Software:** Daniel Domingo-Fernández, Yojana Gadiya, Daniel Rivas-Barragan.

**Supervision:** Daniel Domingo-Fernández, Chris W. Diana, David Healey, Joe Rokicki, Viswa Colluru.

**Validation:** Daniel Domingo-Fernández, Yojana Gadiya, Biswapriya B. Misra.

**Visualization:** Daniel Domingo-Fernández, Yojana Gadiya, Chris W. Diana.

**Writing – original draft:** Daniel Domingo-Fernández, Sarah Mubeen, Biswapriya B. Misra, David Healey.

**Writing – review & editing:** Daniel Domingo-Fernández, Sarah Mubeen, Daniel Rivas-Barragan, Biswapriya B. Misra, David Healey, Viswa Colluru.

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
