## [Decision Letter · Decision Letter 0]

29 Nov 2021

Dear Mr Domingo-Fernández,

Thank you very much for submitting your manuscript "Causal reasoning over knowledge graphs leveraging drug-perturbed and disease-specific transcriptomic signatures for drug discovery" for consideration at PLOS Computational Biology.

As with all papers reviewed by the journal, your manuscript was reviewed by members of the editorial board and by several independent reviewers. In light of the reviews (below this email), we would like to invite the resubmission of a significantly-revised version that takes into account the reviewers' comments.

As you will see from the reviews, both reviewers, but especially Reviewer 2, have considerable concerns regarding the quality of the validation of your methods. They make several suggestions how this could be improved, and we would recommend carefully considering these options. Also, several points of the procedure appear to be somewhat arbitrary (cutoffs, etc...). These should be extensively explained and justified.

We cannot make any decision about publication until we have seen the revised manuscript and your response to the reviewers' comments. Your revised manuscript is also likely to be sent to reviewers for further evaluation.

Sincerely,

Carl Herrmann, Ph.D.

Associate Editor

PLOS Computational Biology

Feilim Mac Gabhann

Editor-in-Chief

PLOS Computational Biology

Reviewer's Responses to Questions

**Comments to the Authors:**

Reviewer #1: This paper describes a novel algorithm that overlays knowledge graphs with drug-perturbed signatures and disease-specific signatures as constructed from 4 data sources. The description of the four data sources can be extended: it is important to know how many drugs, diseases and genes are covered and how much overlap between them (and the KG).

The validation section leaves space for a number of questions and unclarities that need to be addressed. In the description of the algorithm two assumptions/restrictions are described. Paths with cycles or length > 7 are removed. How many are there found? Furthermore, one error in the path is permitted. What does this mean? What is an error? An inconsistent edge? Again, how often does this occur?

For the validation drug-disease pairs are obtained from ClinicalTrials.gov. They are considered as positive labels, even if the ClinicalTrial was aborted? What about negative labels? Did you consider to use a set as a proxy to negative cases? How many drug-disease pairs from CT remained after the harmonisation step? How many of those pairs were found in the prioritisation step (correlation with drug-perturbed signatures and disease signatures)?

Do I read well from table 1 (looking at the TP+FP) that the number of drug-disease pairs validated was resp. 5, 11, 2 and 2? If that is the case this validation is quite limited. If this is not the case more explanation is needed to understand the figures. Why not running all drug-disease pairs from CT against the different sets?

For the interpretation task tow drugs are selected: ponatinib and bicalutamide. How and why have these two drugs selected?

For the target prioritization task it would be better to show the results for all protein target-disease pairs for both Custom and OpenBiolink KG. It is not clear what determines which KG is selected.

Figure 4 is unreadable for me. Too small to see what texts are hidden in the Sankey figure. In the text I think erroneously 4a was referenced for the potatinib case (should be 4b) and 4b for the bicalutamide case (should be 4c).

Baseline: the approach is nice and interesting, the validation needs more attention and requires also a better understanding of the selections/choices made.

Reviewer #2: Review PCB 2021

Domingo-Fernandez et al present a method to prioritise drugs by combining knowledge graphs (KGs), and transcriptomics profiles. The motivation and background are very reasonable, and authors outline and the paper does a good job at pointing directly at the shortcomings we have to deal with at the state of our current knowledge. However, we found that the work dies not find good solutions to actually address them and both the implementation as well as the paper in itself are quite shallow, as we outline below. It might well be that the exiting knowledge and data we have is not ready to be used in this way.

Major comments:

1- the method is based on a combination of 3 ‘step’s in analysis and feels like it i mixing three very different things that might make sense by themselves and ‘see what happens’.

First, finding pathways in KGs seems sensible for a drug, but then overlying transcirptomics of that drug on those pathways is much less clear - the authors argue:

“the observed transcriptomic signatures for proteins in the KG should be concordant with the inferred up- or down-regulations at every step of the path”.

What is the justification for this? e.g. they elaborate: “we expect the inhibition of the protein target” …. Can authors prove this on the datasets they use?

Indeed, the effect of a drug does not need to affect the genes on the same pathway, and that is in fact the fundament of the methods based on ‘causal reasoning’ .

And the comes yet another overlay of transcriptomics in step 3, now from the disease, but only ’on the concordant paths from the previous step (if any)’

2. Further, a fundamental limitation of mapping gene expression to KGs is that the level where authors have “causal” information (proteins) is largely detached from the levels at which they have measurements (RNA). Can authors run their methods using proteomic data to be closer to the actual KGs?

Of note, even if they had adequate protein-level measurements, their method would still only be applicable to the drugs whose effects remain on the protein level exclusively, which not even their selected examples do.

3. Authors perform a number of seemingly arbitrarily methodological choices that need justification (there is some vague general statements in the discussion):

3.1. - Keeping this in mind, a cut-off is applied to each measured gene in the experimental dataset based on the fold change

Which threshold was used and how chosen?

3.2. - Paths with cycles or a length greater than 7 edges between a given drug and disease are not considered,

Why exactly 7?

3.3… - Secondly, we allow for at most one error between the transcriptomic data and a given path

Why 1?

3.4.why 'non-concordant paths are removed.’ In step 2?

4. Benchmarking. Authors benchmarked RPath against 11 equivalent approaches that can be used to prioritize drug-disease pairs based solely on network structure,

Why not methods that use gene expression? It is not fair to compare against methods that use different input data.

5. Validation and overstating results.

5.1. Calling finding ‘ a larger proportion of positive pairs than expected on average by chance.’ a validation is over-selling the results.

5.2. Evaluating based on literature evidence”, seems that they took the output of their prioritisation run and looked up the proteins in PubMed? If so, they are validating their KG by the very data that the KG very likely was built from, and so finding papers in favour of a connection of their prioritised drugs with the disease is not surprising at all. They don’t even attempt to address any confounding of this kind or research bias in general. The discussion also does not mention it.

6. Lack of enough description, in part relating to the specific points above:

6.1. Section 2.3.1 lacks fundamental description of which diseases they are able to look at in the model, and the remainder of the paper also does not mention the pool of diseases and drugs studied.

6.2. Section 2.3.3 argues at length why they can’t apply standard metrics (class label imbalance etc), but they don’t describe their chosen method. BTW, AUC-PR is not an ROC curve, as their explanation suggests.

6.3. Same can be said for the results, presenting 4/5 as “80%” without a measure of confidence.

6.4. Pharmacologically, not explaining the “hits” is underwhelming. Which drug, which disease, does the mechanism of action make sense? They even mention mechanism of action in their abstract…

6.5. The chosen examples look cherry-picked and are not representative of the general notion of “we predict drug mechanisms”, because both are cancer drugs. Both also exemplify the detachment of inference and measurement, as both have extensive transcriptional effects that are not accounted for in the KG (androgen receptor…).

Based on the points above, authors should tone down their claims.

Minor points

1. - The link to the Jupyter notebooks https://github.com/enveda/RPath/tree/master/src/notebooks is broken

2 - “we harmonized these identifiers to the ontologies used in the KG […] using proprietary harmonization scripts”

Disappointing that this part of the work can not be checked as proprietary and lacks reproducibly and transparency

**Have the authors made all data and (if applicable) computational code underlying the findings in their manuscript fully available?**

Reviewer #1: Yes

Reviewer #2: **No: **links to code is broken. harmonization is based on 'proprietary software'

PLOS authors have the option to publish the peer review history of their article (what does this mean?). If published, this will include your full peer review and any attached files.

Reviewer #1: **Yes: **Erik M. van Mulligen

Reviewer #2: No
---

## [Decision Letter · Decision Letter 1]

17 Jan 2022

Dear Mr Domingo-Fernández,

Thank you very much for submitting your manuscript "Causal reasoning over knowledge graphs leveraging drug-perturbed and disease-specific transcriptomic signatures for drug discovery" for consideration at PLOS Computational Biology.

As with all papers reviewed by the journal, your manuscript was reviewed by members of the editorial board and by several independent reviewers. In light of the reviews (below this email), we would like to invite the resubmission of a significantly-revised version that takes into account the reviewers' comments.

While Reviewer 1 is satisfied with your response, Reviewer 2 still has strong reservations, and feels that you failed to adequately address some of their previous comments. We would encourage you to carefully read the comments and address them in a revised version. In particular, you should consider the following points and make appropriate changes:

While I understand that the inclusion of proteomic data goes beyond the scope of your methods, you should at least acknowledge this resource and the fact that corresponding proteomic data is increasingly becoming available, for example through the database mentioned.Given the number of parameters used in your method, it is indeed questionable how they impact the results. While a full exploration of the parameter space is not realistic, I would ask you to provide evidence how changing the number of allowed errors impacts the results, e.g. as a supplementary table.Regarding the comparison with other methods: No alternative method will exactly fit the one developed by the authors. The authors have shown in their comparison that including transcriptomic data in the context of KG improves compared to using the network topology alone. They should however also address the complementary question, namely how using KG together with expression signatures improves over transcriptomic-only approaches for the prediction of drug-disease pairs. In particular, numerous studies have tackled the problem of drug repurposing using transcriptomic signatures.  We would urge the authors to at least discuss in depth those methods, and if possible, provide a comparison with some of these transcriptomic-only methods.Regarding the precision of the method and the comparison with the random null hypothesis: the small number of cases makes the significance of the precision indeed questionable. An alternative null hypothesis should take into account the number of drug diseases-pairs that are already connected in the KG, and not all drug x disease pairs, as the algorithm already selects for connected pairs.I would urge the authors to provide the full list of 30 drug-disease pairs, and discuss these identified hits, to support the claim of generality beyond the two examples already discussed.Please consider uploading the supplementary files to zenodo and providing the corresponding doi as an alternative to github, which should be restricted to providing the code.

As previously mentioned, I would strongly recommend considering the other issues raised by reviewer 2 beyond those listed above.

We cannot make any decision about publication until we have seen the revised manuscript and your response to the reviewers' comments. Your revised manuscript is also likely to be sent to reviewers for further evaluation.

Sincerely,

Carl Herrmann, Ph.D.

Associate Editor

PLOS Computational Biology

Feilim Mac Gabhann

Editor-in-Chief

PLOS Computational Biology

Reviewer's Responses to Questions

**Comments to the Authors:**

Reviewer #1: Thanks for the response on my previous remarks. Nice manuscript that deserves to be read and cited a lot.

Reviewer #2: We acknowledge the efforts of the authors to address our points, but we feel that they could and should have put more effort in doing so. Some comments are simply dismissed, as showed below. The fact that the revised version was sent back in only a few days suggests that indeed not too much effort was put in revising the manuscript.

As bottom line, the paper tries to introduce KG-based learning as drug prioritisation tool, but fails to convince us that this can be actually useful, given the current state of our knowledge and the algorithm at hand. The paper for us is too detached from clinical reality and does not try to get into the details of how a drug works and why this could be mirrored by their algorithm. If the “validations” are not significantly extended, we do not think that one should believe that these results can explain drug mechanisms of action and prioritise drugs for new diseases.

Below some comments on the answers to our points:

# Some questions have been misunderstood it seems and thus the answer does not relate to the question. For example to the question:

Indeed, the effect of a drug does not need to affect the genes on the same pathway, and that is in fact the fundament of the methods based on ‘causal reasoning’. And then comes yet another overlay of transcriptomics in step 3, now from the disease, but only ’on the concordant paths from the previous step (if any)’

Authors answered:

* Although the effect of a drug does not always have to affect the genes on the targeted path, it is reasonable to think that for the majority of the cases, an effective drug for a given disease will likely alter the signatures of the targeted path. The idea of overlaying transcriptomic signatures from diseases and drugs is widely established and it is the basis of well-known drug discovery libraries such as CMap and L1000.

But The paradigm is to match indeed signature, but that was not the question, rather why matching the expression to the path where the drug targets.

# An example of how authors are dismissive about our comments is the question about proteomics vs transcriptomics, which they answer with factually wrong answers:

1 Proteomics data does not yet have the same coverage as RNA-seq or other gene expression techniques that we have used in our study (thousands of different transcripts are measured vs. a few proteins).

Mass-spectrometry proteomics provides coverage of thousands of proteins. It might not be genome wide, but there is enough coverage to at least try.

1. As of yet, there are no major repositories of proteomic datasets as there are for gene expression (e.g., GEO or ArrayExpress).  

This is incorrect. See e.g. PRIDE, https://www.ebi.ac.uk/pride/archive/

# Methodological choices of e.g. parameters should have bene further justified or at least explored the implications of changing them. For example,

3.3... - Secondly, we allow for at most one error between the transcriptomic data and a given path. Why 1?

Using zero as a threshold, there were no drug-disease pairs prioritized because the chance of finding a perfect concordant path is extremely low (see reasons in the second paragraph in the discussion section). Thus, we decided to allow for at most one error, precluding the possibility also of too high a leniency.

This seems an arbitrary decision, why not exploring e.g. 2, 3,…?

# Comparisons to other methods should be further elaborated:

As per our knowledge, there are no network-based methods that operate on multimodal KGs using both transcriptomics modalities and that also have software available that can be used to prioritize drug-disease pairs.

Can authors not use any network base method for gene expression that was perhaps not developed for KGs but for molecular networks and extend to KGs?

# Some answers are not clear, e.g.

> Nonetheless, because the majority of the interactions in the KG are not relevant to any particular disease condition (except for the last edge in the path connecting a protein to a disease), the interactions encoded in a KG do not directly reflect literature knowledge, nor favour connections of prioritized drugs with the disease.

This is a confusing statement “KG do not directly reflect literature knowledge,” we thought they do reflect literature knowledge, that is the nature of a knowledge graph?

#validations are still unconvincing in their scope:

> As mentioned in the previous point, we used precision as a metric. In this case, 5 drug-disease pairs were prioritized and among them 4 have been clinically investigated, resulting in the reported precision of 80%. Furthermore, we demonstrate in several datasets and two different KGs that, nearly across the board, the precision the algorithm achieves is better than the expected precision by chance (Table 1).

In a small set of 5 “validators”, the chance for spuriously high precision is quite significant. Considering that the algorithm uses gene-disease relationships (final connection) and allows only one mismatch, it effectively filters for a set of paths that have higher likelihood of being disease-related already. That the precision here is higher than average should be self-explanatory. Combined with the multitude of possibilities given by omics measurements, the chance of spurious association seems too high (see Ioannidis 2005, https://doi.org/10.1371/journal.pmed.0020124).

> There are 30 prioritized pairs whose investigation would require a considerable manual effort. Thus, we focused extensively on the proposed mechanism of actions of only two of these drugs in the Interpretation section 2.2.

Since the authors claim generality and also acknowledge that there is no baseline to compare their method against (see above), they should supply proof of this generality. Otherwise, at least tone down the claims.

For the reader, it is not apparent whether the chosen two results are representative or cherry-picked. This is of particular relevance because the abstract states that "The paths which match this signature profile are then proposed to represent the mechanism of action of the drug.”

For someone knowledgeable in pharmacology, it is not a lot of effort to sanity-check 30 predictions. Also, why can the results not be given as a supplementary table, such that the knowledgeable reader can form their own opinion of how meaningful are these predictions?

For instance, given the now-available pool of drugs and diseases, we would be interested on whether glucocorticoids or 4-ASA show up in the treatment of inflammatory bowel disease or ulcerative colitis. Antidiabetics in diabetes? It is also intriguing that psychiatric diseases are in the list, while psychiatric medications are quite underrepresented in the pool. Instead, there are several anti-infectives, which do not target the human organism at all; and cancers are severely overrepresented as well.

Minor:

# Provide all information in a long-term manner:

> We have now added a file which displays the drugs and disease with respect to their source data annotation on GitHub (https://github.com/enveda/RPath/blob/master/data/drug_disease_overview.tsv). This file is now referenced in the Supplementary Information.

This is not adequate to point to a repository; a manuscript should contain all information that is required for the reader to understand capabilities and implications of the algorithm/method. Additionally, GitHub is not a permanent storage facility, authors should deposit everything relevant for the paper in e.g. zenodo.

* For the sake of reproducibility and transparency of the paper, we released the mappings corresponding to the diseases analyzed in the paper. The mappings are available at https://github.com/enveda/RPath/blob/master/data/manual_drug_disease_mapping.tsv and the notebooks have been accordingly updated. The corresponding sentence has been modified accordingly.

Are the ‘proprietary harmonisation scripts’ also provided or only the results mapping? Otherwise, can the tool be used by others in other contexts?

**Have the authors made all data and (if applicable) computational code underlying the findings in their manuscript fully available?**

Reviewer #1: Yes

Reviewer #2: **No: **is unclear of code is available or only resulting mappings. also as noted some data is in a github repository so could get lost or hidden again.

PLOS authors have the option to publish the peer review history of their article (what does this mean?). If published, this will include your full peer review and any attached files.

Reviewer #1: **Yes: **Erik M. van Mulligen

Reviewer #2: No
---

## [Editor Report · Decision Letter 2]

9 Feb 2022

Dear Mr Domingo-Fernández,

We are pleased to inform you that your manuscript 'Causal reasoning over knowledge graphs leveraging drug-perturbed and disease-specific transcriptomic signatures for drug discovery' has been provisionally accepted for publication in PLOS Computational Biology.

Best regards,

Carl Herrmann, Ph.D.

Associate Editor

PLOS Computational Biology

Feilim Mac Gabhann

Editor-in-Chief

PLOS Computational Biology

---

## [Editor Report · Acceptance letter]

22 Feb 2022

PCOMPBIOL-D-21-01914R2 

Causal reasoning over knowledge graphs leveraging drug-perturbed and disease-specific transcriptomic signatures for drug discovery

Dear Dr Domingo-Fernández,

I am pleased to inform you that your manuscript has been formally accepted for publication in PLOS Computational Biology. Your manuscript is now with our production department and you will be notified of the publication date in due course.

With kind regards,

Zsanett Szabo
